# Attitudes of Farmers and Rural Area Residents Toward Climate Change Adaptation Measures: Their Preferences and Determinants of Their Attitudes

**Kenshi Baba** [1,*] **and Mitsuru Tanaka** [2]

1   Faculty of Environmental Studies, Tokyo City University, Tokyo 224-8551, Japan
2   Faculty of Social Science, Hosei University, Tokyo 194-0298, Japan; mtanaka@hosei.ac.jp
*   Correspondence: kbaba@tcu.ac.jp; Tel.: +81-45-910-2554

**Abstract:** In this study, data obtained from an online survey were analyzed to identify the perception gap between farmers and nonfarmers (rural area residents) toward climate change adaptation measures with conventional and new elements of the psychological mechanism. Key findings from the study were as follows. First, the perception of climate change risk and awareness of impacts of climate change had strong effects on the preferences for and willingness to participate in measures rather than trusting the government and values pertaining to the policy decision-making process. Second, farmers tended to prefer "protection" and "transfer of risks (insurance)" as climate change adaptation measures more than nonfarmers did. Farmers also tended to be unwilling to participate in "withdrawal", reflecting the difficulty of relocating agricultural land. Third, farmers' willingness to participate in climate change adaptation measures was determined strongly by their preferences. Therefore, to increase preference, there needs to be communication about multiple risks including climate change risks associated with not only "adjustment" and "protection", which tend to be preferred, but also "withdrawal", which tends to not be preferred. Contrasting with these, nonfarmers tended to prefer any particular climate change adaptation measures statistically-significantly, but they tended to be willing to accept "self-help" absolutely and "withdrawal" relatively. Also, farmers' willingness to participate in climate change adaptation measures was determined strongly by their preference. One of the ways to increase the preference is communicating about the multiple risks including climate change risks associated with "adjustment," "protection" and "transfer" which tend to be preferred more than nonfarmers did. Finally, trust in the government and values pertaining to the policy decision-making process did not necessarily have a serious impact on policy preferences and willingness to participate, both for farmers and nonfarmers. More analyses for other sectors will be needed for further study.

**Keywords:** climate change risk; adaptation policy; risk perception; agriculture; online survey

## 1. Introduction

The Intergovernmental Panel on Climate Change (IPCC) published its Fifth Assessment Report [1], which left no doubt about the reality of global warming. The report demonstrated that climate change had made "impacts on water resources (water volume and quality)", "changes to habitats for land, freshwater, and marine organisms", and "impacts on crops", and it predicted eight key future risks: rising sea levels and high tides, floods and heavy rainfalls, infrastructure breakdown, heat-stroke, food shortages, water scarcity, loss of marine ecosystems, and loss of land ecosystems. The report

then proposed to advance both adaptive and mitigating measures to manage these future risks and to promote the realization of a resilient society and sustainable development.

The impacts of climate change on agriculture are serious and pressing concerns. Since 2008, the Ministry of Agriculture, Forestry and Fisheries of Japan (MAFFJ) has been conducting a fact-finding survey on problems believed to be impacts of climate change such as high-temperature stress observed at agricultural production sites and on adaptive measures for such impacts. The Ministry released the results in the "Annual Survey Report on the Impacts of Global Warming [2]", which revealed that there has been an ongoing trend toward higher temperatures. Adaptation measures undertaken to address this problem include the introduction of heat-tolerant rice varieties in combination with cultivation techniques, such as late planting and water management in rice farming, as well as the introduction of fruit varieties with superior coloring in combination with reflective film materials and other techniques in fruit cultivation. In addition, in November 2015, the Cabinet of Japan approved the national government's National Plan for Adaptation to the Impacts of Climate Change, and, internationally, the Paris Agreement was signed, committing to keep the average temperature rise prior to the Industrial Revolution to under two degrees Celsius. By issuing these plans, the Japanese government has set forth its adaptation strategies at the national level.

With respect to the implementation of adaptation measures, many studies on climate change adaptation in general terms have pointed out the need to promote communication and collaboration among experts, stakeholders, and the general public to integrate their own knowledge in order to conduct risk and vulnerability assessments at the local level (e.g., Laukkonen et al. [3] and Halsnæs et al. [4]). In particular, to improve the acceptability of adaptation policies, efforts to resolve the potential perception gap between policymakers/experts and the public about the impacts and risks of climate change and to seek the public's understanding and cooperation in implementing the policies are essential (e.g., van Aalst et al. [5]). Especially for the different perceptions and attitudes of stakeholders toward climate change adaptation, various methodologies of consensus-building and risk communication have been proposed (e.g., Otto-Banaszak et al. [6] and Baba et al. [7]).

For the agricultural sector in Japan, analysis has been conducted on the perceptions of experts (Suda et al. [8]) and stakeholders (Fujisawa and Kobayashi [9] and Matsuura et al. [10]). However, few studies have been carried out which clarify the farmers' psychological mechanisms for their perception of climate change risk and attitudes toward adaptation measures in Japan. Meanwhile lack of information on adaptation measures and socioeconomic constraints are the main barriers to adopt adaptation measures for farmers around the world (e.g., Bryan et al. [11], Deressa et al. [12], Below et al. [13], Abid, et al. [14], Acquah [15], Dumenu et al. [16], Kibue et al. [17], and Belay et al. [18]). Other studies on farmers' psychological mechanisms demonstrated that farmers regarded climate change as high risk and would be more likely to have an intention of adaptation than those who saw the events as part of normal variation (e.g., Gordon et al. [19], Dang et al. [20], and Carlton et al. [21]). The most important determinants of adaptation were farmers' perceptions of risk to their own farms and attitudes toward innovation and adaptation measures (Mase et al. [22]).

As mentioned above, understanding the perception gap between farmers and rural area residents to climate change risk and adaptation measures is key for policy implementation. To this end, the authors are developing a trans-interdisciplinary approach, applying it to Nagano prefecture in Japan, and then developing a narrative scenario about climate change impacts on agriculture and daily life in the near future. To help develop this scenario, the purpose of this study was to identify the perception gap between farmers and nonfarmers (rural area residents) toward climate change adaptation measures with conventional and new elements of the psychological mechanism. These findings will tell us how to promote greater acceptance of adaptation measures in the agricultural sector.

## 2. Research Methods

Based on these previous studies, to clarify differences on how the psychological mechanism constitutes the determinants of attitudes for climate change adaptation measures between farmers

and nonfarmers (rural area residents), a questionnaire survey was conducted, as described in Table 1. As mentioned above, Nagano prefecture was selected for the questionnaire survey to help develop the scenario. Nagano prefecture has the second highest production volume of both apples and grapes in Japan (142,100 tons with 7560 ha of planted area in 2016 for apples and 28,800 tons with 2300 ha of planted area in 2016 for grapes), but projections tell us that Nagano will no longer be a major production area for apples and will be more suitable for grapes by the end of this century at the latest. This is why the authors are developing a narrative scenario for climate change impact in Nagano. Also, two more areas were selected for the questionnaire survey: Aomori prefecture, which has the highest production volume of apples in Japan (447,800 tons with 19,900 ha of planted area in 2016 for apples), and Yamanashi prefecture, which has the highest production volume of grapes in Japan (42,500 tons with 3860 ha of planted area in 2016 for grapes).

Survey respondents were recruited by an online survey company. Nonfarmers (rural area residents) living in the above mentioned three prefectures were sampled from ten municipalities in Aomori prefecture with particularly high production volumes of apples (Hirosaki City, Aomori City, Hirakawa City, Kuroishi City, Itayanagi Town, Nanbu Town, Goshogawara City, Tsuruta Town, Owani Town, and Fujisaki Town); five municipalities in Yamanashi Prefecture with particularly large production volumes of grapes (Fuefuki City, Koshu City, Yamanashi City, Minami-Alps City, and Kofu City); and six cities in Nagano Prefecture with particularly large production volumes of apples and grapes (Nagano City, Matsumoto City, Suzaka City, Nakano City, Azumino City, and Shiojiri City). People living in these areas were distributed by age (20s–30s, 40s, and 50s–60s) and gender into six categories (three age groups × two genders in each prefecture). The number of respondents in each category was approximately 35 (N = 623). Also, survey respondents of farmers were recruited by an online survey company from all over the country. A screening survey was conducted among farmers in advance to select respondents who were full-time farmers, part-time class 1 farmers, or part-time class 2 farmers primarily growing crops rather than livestock (N = 412). A comparison of farmers and nonfarmers in the same region would have been ideal, but this idea was abandoned because it was difficult to secure samples. The details of the personal data of respondents remained secure by the survey company with the contract in which the data were used within the scope of the purposes in principle.

**Table 1.** Overview of survey.

| Survey Period | 11–13 March 2014 |
|---|---|
| Respondents | Farmers across Japan and people who are not engaged in farming living in Aomori prefecture, Yamanashi prefecture, and Nagano prefecture (recruited by a survey company). |
| Survey method | Requests were sent by email and respondents filled out the questionnaire on a website. |
| Questions | Risk perception and awareness of climate change, preferences for and willingness to participate in adaptation measures, trust in the government, values about policy decision-making processes, individual attributes, etc. |
| Distribution | • Farmers: A screening survey was conducted among farmers from all over the country in advance to select respondents who were full-time farmers, Part-time Class 1 farmers, or Part-time Class 2 farmers with a main crop other than livestock. <br> • Nonfarmers: Requests were sent so as to ensure even distribution of respondents by gender and age in the three prefectures (respondents were distributed into six categories by age (20s–30s, 40s, and 50s–60s) and genders in each prefecture). |
| N of Response | farmers: 412, nonfarmers (rural area residents): 623 |

Questions were roughly grouped into the following: risk perception and awareness of climate change, preferences for and willingness to participate in adaptation measures, trust in the government, values with respect to policy decision-making processes, and individual attributes. Most of these were conventional elements in the psychological mechanism, which was utilized in the above mentioned previous studies. Trust was introduced as a new element, which had not been introduced so far in this

context. These questions were modified from the ones employed in a prior survey on a different theme (Baba et al. [23]) for the purposes of this study. The approaches and intentions behind these questions will be explained below where relevant.

## 3. Results

### 3.1. Each Determinant of Attitude for Climate Change Adaptation Measures

First, the results of each determinant of attitudes for climate change adaptation measures of farmers and nonfarmers (rural area residents) are shown in Figures 1–3. These were the elements that had statistically significant differences observed between farmer and nonfarmers except for the perception of climate change risks.

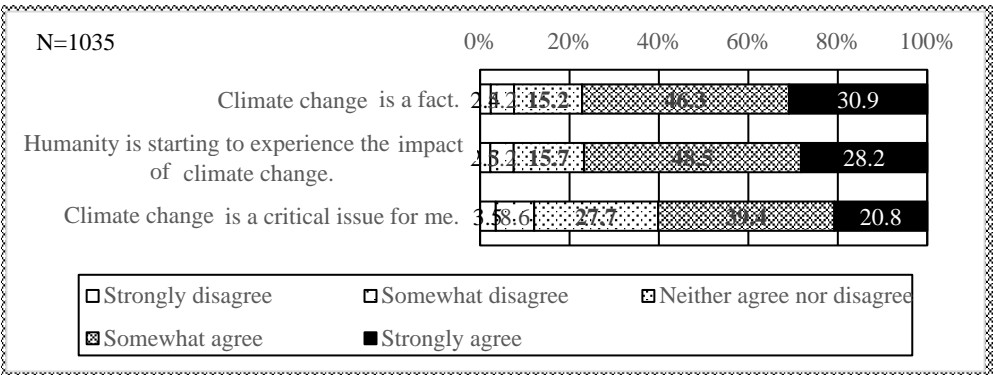

**Figure 1.** Perception of climate change risks.

The respondents' perceptions of climate change risks were measured using three different criteria. The question was "To what extent do you agree or disagree with the following three statements about climate change risk? Please answer with five-point scale." As shown in Figure 1, while approximately 76% of the respondents perceived the risks as objective facts or as threats to humanity as a whole, only approximately 60% perceived the risks at a personal level. No statistically significant differences were observed between farmers and nonfarmers for any criteria.

Figure 2 shows the results of a question concerning the respondents' awareness of the impacts of climate change on their daily lives in the last several years regarding seven criteria that could potentially pose environmental and health risks. The question was "To what extent are you aware of the impacts of climate change in the following seven phenomena? Please answer with five-point scale." Overall, "storms and floods such as localized heavy rainfalls" had the largest percentage of respondents saying they were "aware" ("Very aware": 21.7%, "Somewhat aware": 44.9%). This was followed by "impacts on lifestyle" such as disruption of traffic networks from heavy snow, "heatstroke and other damage to health", and "damage to food production" such as quality degradation and harvest reduction in crops and seafood.

Statistically significant differences were observed between farmer and nonfarmers. The percentage of farmers who were "aware" tended to be significantly higher than the percentage of nonfarmers in most criteria, especially "damage to food production". On the other hand, more non farmers than farmers were "aware" of "impacts on lifestyle", unlike in other criteria. This was likely attributable to the fact that one-third of the nonfarmers were sampled from Yamanashi Prefecture, which suffered damage from heavy snow that hit the region in February 2014, and the farmers were sampled from across the country. However, it was only in this criterion of this question that nonfarmers showed a significant difference among prefectures. No significant difference was observed in many other questions such as preferences for and willingness to participate in adaptation measures. As such, prefectures will not be used as a variable to tabulate the data.

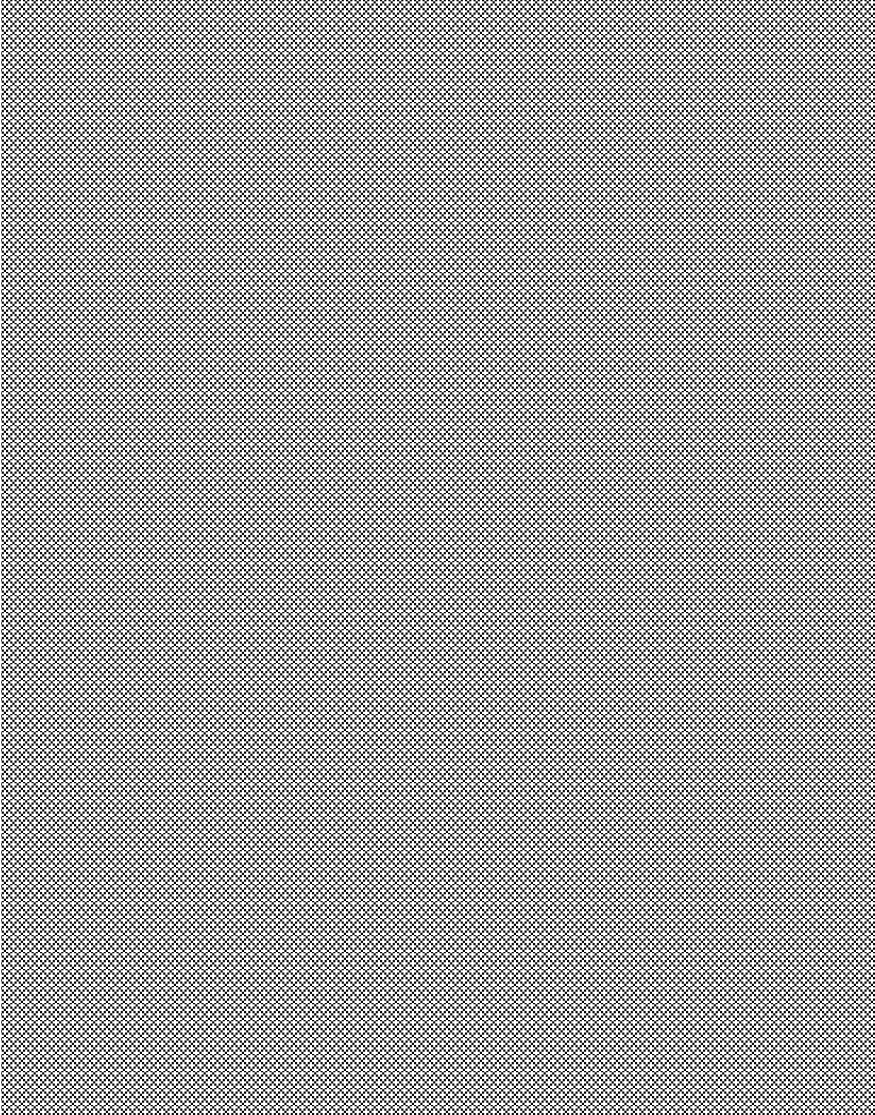

**Figure 2.** Awareness of the impacts of climate change among farmers and nonfarmers. ** $p < 0.01$, * $p < 0.05$, *p*-value: less than 0.05 is interpreted as statistically significant in general.

Drawing on basic criteria of adaptation measures (Mimura [24]) and the criteria of risk measures used in risk management (Yano [25]), five criteria were employed similar to Baba et al. [23]: protection, adjustment, withdrawal, risk transfer (insurance), and risk retention (do nothing). Then, the respondents were asked about their general "preferences" for these measures using a five-point scale. The question was "To what extent do you prefer with regard to the following five types of climate change adaptation measures in general terms? Please answer with five-point scale." As for five adaptation measures, farmers were prepared for agriculture-related measures, and daily life measures were expected to be undertaken in rural areas for the nonfarmers. As such, slightly different expressions were used to describe the criteria. In addition, another five criteria were established with respect to the specific ways in which the respondents were willing to participate in adaptation measures, which were: public aid, self-help, mutual aid, withdrawal, and risk retention (do nothing). The respondents were similarly asked to assess the criteria on a five-point scale. The question was "To what extent are you willing to participate in the following five types of climate change adaptation measures? Please answer with five-point scale."

As shown in Figure 3, relatively large percentages of respondents showed positive attitudes toward "responding to the circumstances in the local community and adapting my lifestyle (adjustment for nonfarmers)/responding to the circumstances in the local climate and adapting my way of farming such as cultivar improvement (adjustment for farmers)" and "protecting housing and infrastructure through facility maintenance (protection for non-farmers)/protecting agricultural infrastructure through facility maintenance (protection for farmers)" as a whole. In contrast, small percentages of respondents showed positive attitudes toward transferring risks by "covering the cost with insurance in case of damage" and risk retention (doing nothing). A share of respondents showed a positive response to withdrawal by "withdrawing from the high-risk area and relocating the house (for nonfarmers)/agricultural land (for farmers) to another area". Hence, the respondents preferred predictable responses, such as adjustments and protection, but they were less receptive to withdrawal and risk transfer.

Meanwhile, relatively large percentages showed a positive attitude toward the adaptation measures of the "responsibility of the central and municipal governments (public aid)", and the percentage of respondents who were positive toward the "independent protective measures (self-help)" was overwhelmingly higher than others. Statistically significant differences were observed between farmers and nonfarmers. With respect to "preferences", farmers preferred "protection" and "transfer of risks (insurance)" more than nonfarmers did. With respect to "willingness to participate", a small percentage of farmers selected self-help and withdrawal.

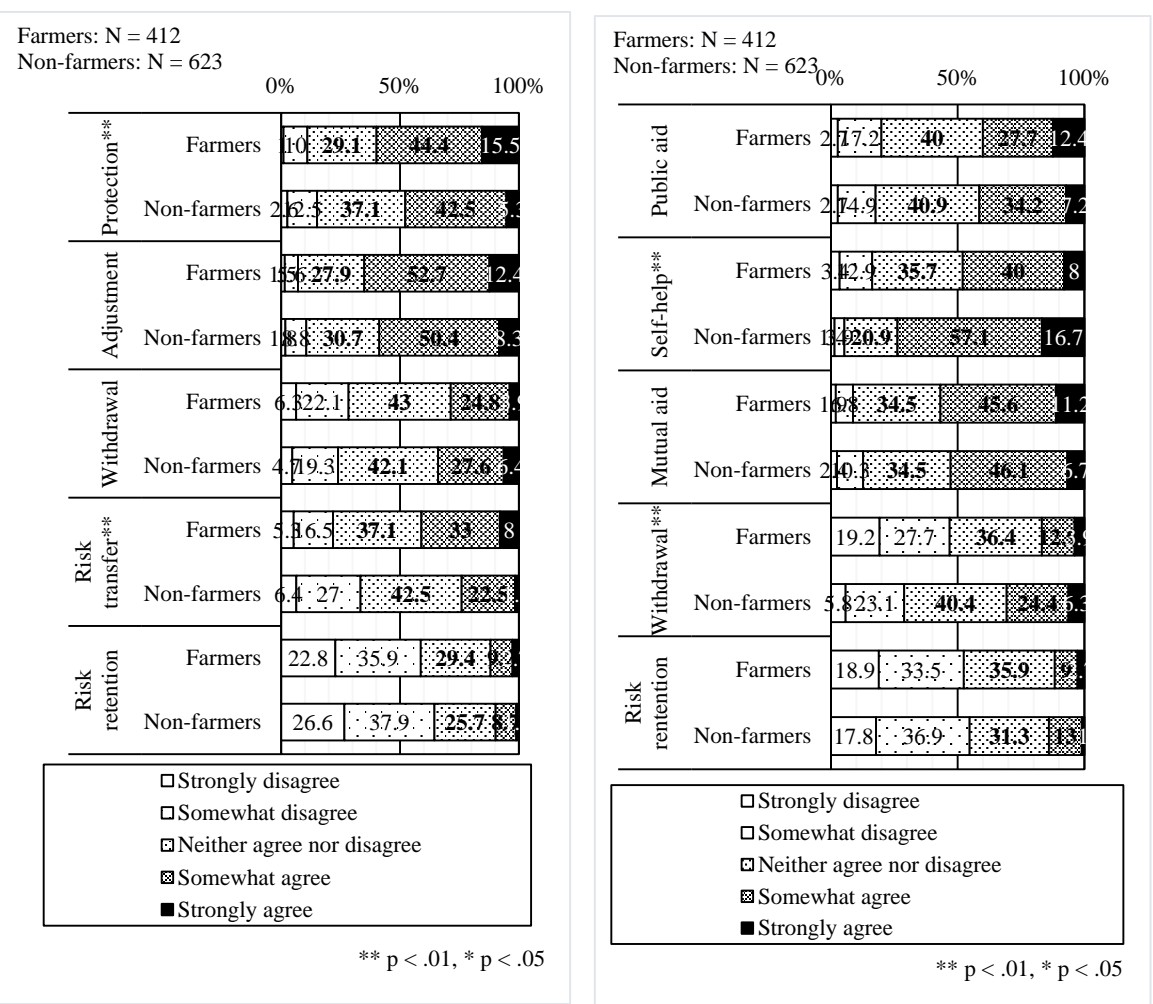

**Figure 3.** Preferences for and willingness to participate in adaptive measures among farmers and nonfarmers. ** $p < 0.01$, * $p < 0.05$, $p$-value: less than 0.05 is interpreted as statistically significant in general.

### 3.2. Psychological Mechanism for Perception of Climate Change Risk and Attitudes toward Adaptation Measures

Based on the results described above, a covariance structure analysis was conducted using conventional elements, such as trust in the government and values with respect to policy decision-making processes, in addition to awareness of the impacts of climate change and perception of climate change risk in order to comprehensively identify determinants of attitudes toward climate change adaptation measures. Trust in the government and values with respect to policy decision-making processes were employed as significant variables in a similar type of model (Baba et al. [23]). The background on the application of these variables is described below.

In general, determinants of the public's acceptance of policies included various procedural aspects of the policy decision-making process such as information disclosure and public participation. Baba [26,27] established seven criteria based on studies by Tyler and Degoy [28], Webler [29], and Renn [30] to measure procedural fairness and distributive fairness that should be emphasized in the context of deciding the locations of new facilities. For the purpose of this study, these criteria were modified to nine criteria that focused on communication among citizens and government authorities/experts in the context of addressing climate change risk, and the respondents were asked about how they valued the criteria using a five-point scale.

In addition, with respect to the meaning of "trust" in the context of risk management, the important issue is whether the public can trust experts and government authorities as risk managers, as it is difficult for the public and experts to hold a technical discussion on an equal basis. Studies conducted so far generally argued that trust was established based on "expectations of intentions" and "expectations of abilities" (Yamagishi [31]). Recently, however, Nakayachi and Cvetkovich [32] suggested that trust was established by the fact that one shared the same values as the other party (salient value similarity). This point was taken into account, and nine criteria regarding trust in local governments were established for this study. Then, the respondents were asked about how they valued the criteria using a five-point scale.

First, according to the calculated Cronbach's alpha, which is one of the reliability coefficients indicating how closely related a set of items are as a group, it was examined whether each criterion had internal consistency as an observable variable for each latent variable (i.e., determined which criteria would be employed for analysis). Then, the models that had high goodness of fit levels (i.e., with no inconsistencies in the signs of coefficients among latent variables) were selected from a number of combinations of variables. As a result, the models shown in Figure 4 were adopted. The goodness of fit indices are shown in Table 2. Based on the estimated standardized coefficients in Figure 4 as a whole, it was demonstrated that the respondents' general "preferences for adaptation measures" had a strong effect on their "willingness to participate in specific adaptive measures". Further, it was also shown that the respondents' "perception of climate change risk" had a relatively strong effect on their "preferences" and "willingness to participate", and that "trust in the government" and "values about policy decision-making processes" had some impact on their willingness to participate.

**Table 2.** Different goodness for fit indices of the results of covariance structure model analysis.

|  | AGFI | CFI | RMSEA | N |
|---|---|---|---|---|
| All | 0.917 | 0.925 | 0.049 | 1035 |
| Farmers | 0.882 | 0.918 | 0.052 | 412 |
| Nonfarmers | 0.900 | 0.922 | 0.051 | 623 |

AGFI: Adjusted Goodness of Fit Index, GFI: Goodness of Fit Index, RMSEA: Root Mean Square Error of Approximation.

Looking at the farmer and nonfarmer distinction in Figure 4 and Table 2, the goodness of fit indices were slightly higher in the nonfarmer model, but they were almost equivalent. The standardizing coefficient of the farmers' policy "preferences", vis-a-vis their "willingness to participate", was higher,

suggesting that farmers' willingness to take a specific action was more dependent on their general preferences for adaptation measures than it was for nonfarmers. On the contrary, the standardizing coefficient of the nonfarmers' "perception of climate change risk", vis-a-vis their "willingness to participate", was higher than farmers, suggesting that nonfarmers' general perception of climate change risks had impact on their willingness to take a specific action as well as their general preferences for adaptation measures.

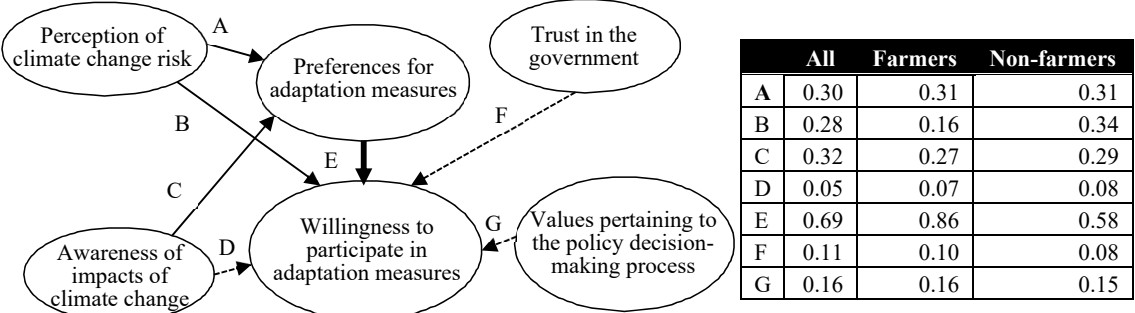

| | **All** | **Farmers** | **Non-farmers** |
|---|---|---|---|
| **A** | 0.30 | 0.31 | 0.31 |
| **B** | 0.28 | 0.16 | 0.34 |
| **C** | 0.32 | 0.27 | 0.29 |
| **D** | 0.05 | 0.07 | 0.08 |
| **E** | 0.69 | 0.86 | 0.58 |
| **F** | 0.11 | 0.10 | 0.08 |
| **G** | 0.16 | 0.16 | 0.15 |

\* All estimated parameter values have a significant level of 1%.

**Figure 4.** Determinants of the attitudes toward climate change adaptation measures (results estimated using covariance structure analysis).

## 4. Discussion

Along with the previously mentioned studies on farmers' psychological mechanisms, which demonstrated that important determinants of adaptation were farmers' perceptions of risk, attitudes toward innovation, and adaptation measures, this study revealed the perception of climate change risk and awareness of impacts of climate change were important to policy preferences and willingness to participate. In addition, let us examine the differences of psychological mechanisms between farmers and nonfarmers.

The percentage of farmers who were aware of climate change tended to be significantly higher than the percentage of nonfarmers in most criteria, particularly regarding "damage to food production", whereas the percentage of farmers who perceived the risks of climate change tended not to be significantly higher than the percentage of nonfarmers. This results from farmers being aware of climate change impacts in daily agricultural practice, so they are particularly sensitive to climate change risks. Also, farmers tended to prefer "protection" and "transfer of risks (insurance)" as climate change adaptation measures more than nonfarmers. This results in that most of them were unwilling to participate in "withdrawal", reflecting the difficulty of relocating agricultural land.

Contrasting with these, nonfarmers tended to prefer any particular climate change adaptation measures statistically-significantly, but they tended to be willing to accept "self-help" absolutely and "withdrawal" relatively. Also, in the case of nonfarmers, risk perception of climate change determined, relatively strongly, willingness to participate in climate change adaptation measures.

Meanwhile, farmers' willingness to participate in climate change adaptation measures was determined strongly by their preference. One of the ways to increase the preference is communicating about the multiple risks including climate change risks associated with "adjustment," "protection" and "transfer" which tend to be preferred more than nonfarmers did.

Trust in the government and values pertaining to the policy decision-making process, which were new elements prepared in the psychological mechanism, did not necessarily have a serious impact on policy preference and willingness to participate, both for farmers and nonfarmers. In the context of risk management, the important issue is whether the public can trust experts and government authorities as risk managers, as it is difficult for the public and experts to hold a technical discussion on an equal basis. In the context of the agricultural sector in climate change, the perception of climate change risk and awareness of impacts of climate change were more important. However, this does not mean that

trust is not important in the context of climate change. As Baba et al. [23] indicated in the context of the disaster reduction sector in climate change, values pertaining to the policy decision-making process were important for policy preferences and willingness to participate as well as for the perception of risk and benefit. As climate change adaptation is a cross-sectoral issue, the psychological mechanism for perception of climate change risk and attitude to adaptation measures is expected to differ depending on the sectors.

## 5. Conclusions

In this study, data obtained from an online survey were analyzed to identify the perception gap between farmers and nonfarmers (rural area residents) toward climate change adaptation measures with conventional and new elements of the psychological mechanism. The findings from the study are summarized below.

First, along with the previously mentioned studies of farmers' psychological mechanisms, this study revealed that perception of climate change risk and awareness of impacts of climate change were important for policy preferences and willingness to participate.

Second, the farmers were aware of climate change impacts in daily agricultural practice, so they were particularly sensitive to climate change risks. Also, farmers tended to prefer "protection" and "transfer of risks (insurance)" as climate change adaptation measures more than nonfarmers did. This resulted in that most of them were unwilling to participate in "withdrawal", reflecting the difficulty of relocating agricultural land.

Third, farmers' willingness to participate in climate change adaptation measures was determined strongly by their preference. One of the ways to increase the preference is communicating about the multiple risks including climate change risks associated with "adjustment," "protection" and "transfer" which tend to be preferred more than nonfarmers did. Finally, trust in the government and values pertaining to the policy decision-making process did not necessarily have a serious impact on policy preference and willingness to participate, both for farmers and nonfarmers. As climate change adaptation is a cross-sectoral issue, the psychological mechanism for attitude to adaptation measures is expected to differ depending on the sectors. Other analyses in other sectors will be needed for further study. By doing this, we can gain the insight of foster public acceptance of climate change adaptation measures in each sector.

**Author Contributions:** Conceptualization, K.B. and M.T.; Methodology, K.B.; Software, K.B.; Validation, K.B. and M.T.; Formal Analysis, K.B.; Investigation, K.B.; Resources, K.B.; Data Curation, K.B.; Writing-Original Draft Preparation, K.B.; Writing-Review & Editing, K.B. and M.T.; Visualization, K.B.; Supervision, M.T.; Project Administration, M.T.; Funding Acquisition, K.B. and M.T.

**Funding:** This study was supported by the Environment Research and Technology Development Fund (S-8) of the MOE (the Ministry of the Environment) of Japan, a Grant-in-Aid for Scientific Research (C) (No. 26340122) of the JSPS (Japan Society for the Promotion of Science), and by SI-CAT (Social Implementation Program on Climate Change Adaptation Technology) of the MEXT (the Ministry of Education, Culture, Sports, Science and Technology) of Japan.

**Acknowledgments:** The authors would like to express sincere gratitude to the respondents who participated in the questionnaire survey, to Motoko Kosugi (Graduate School of Integrated Science and Technology, Shizuoka University) who collaborated with us especially in designing the questionnaire, to Yuko Kawai (the former graduate student, Graduate School of Public Policy, University of Tokyo), Ayami Akagawa (Graduate School of Design, Harvard University), Izumi Hirata (Graduate School of Engineering, University of Tokyo), and Eri Amanuma (Graduate School of Agricultural and life sciences, University of Tokyo) who supported our research activity.

**Conflicts of Interest:** The authors declare no conflict of interest. The funders had no role in the design of the study; in the collection, analyses, or interpretation of data; in the writing of the manuscript, or in the decision to publish the results.

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
