# Peer review of "Attitudes of Farmers and Rural Area Residents Toward Climate Change Adaptation Measures: Their Preferences and Determinants of Their Attitudes"

_climate, doi:10.3390/cli7050071_

Round 1

Reviewer 1 Report

None

Author Response

Just upload the revised version.

Reviewer 2 Report

This manuscript describes a basic survey of the attitudes of rural area residents of Nagano prefecture in Japan towards climate change adaptation measures with emphasis on the differences between farmers and non-farmers. It falls into the scope of the journal.

When reading this article for the first time my first thoughts were that this is yet another survey of attitudes towards climate change and a variety of adaptation measures. Looking back over my paper review records I have reviewed over 20 such manuscripts in the last 6 months across several different journals. I was hoping that this one would offer something new but unfortunately this is not the case.

This led me to consider the rationale for the work which, in conclusion, is severely lacking. The article needs to explain exactly what the value of the study is, where it adds and extends existing knowledge and what the benefits are likely to be. It also needs to explain why the findings of the multitude of existing studies cannot be applied to this particular location. The authors do include a section on previous studies which is very useful and state that no large-scale survey’s have been conducted previously. This is not the case. A quick look through my own records and a search with Google scholar revealed articles on several large scale surveys some larger than that described in this article. When the lack of a sound rationale is considered along with the findings of other studies in the context of the findings of this particular study I cannot see what new information has been identified.

My second major comment is in respect to the structure of the article which is somewhat muddled. I would strongly advise that that the section on previous studies is included into the introduction rather than with the research methods section. The introduction should provide background information (which it does), provide an overview of the current state of the science base (i.e. describe other studies) and use this to lead into the rationale for the work. The section on the research methodology should have clean and concise sections on the location, how the survey was conducted (including exactly how respondents were selected, how many, criteria for selection etc.), details of the survey and how it was analysed. Much of this information is provided but not in discrete sub-sections. The results section is adequate although it does tend to repeat a lot of information that is evident from the figures. The text should rather try and draw out information that is not so obvious. There is also a lot of text in the results section that would be better placed in the discussion section such as that pertaining to ‘trust’. The conclusion section currently repeats much of what has been said in the results section and does not adequately reflect back on the studies aims and objectives. The conclusion should also make it clear what new science has emerged from the work and how this adds to previous studies.

A few further comments:

- The quality of the tables, figures and diagrams is poor. These need to be re-done.

- Much of the text is written in the first person (i.e. using ‘we’). This has a tendency to focus the readers attention on the researchers and not on the scientific content. The latter being the purpose of publication. Please depersonalise the entire manuscript.

- I would also suggest the use of the English language is checked by a native English speaker. It is not poor but rather clumsy in a few places.

Author Response

For the first major comment, we examined previous studies again, added some papers, summarized again at the new introduction section, which moved from the former previous studies and research methods section, and described our new information at the discussion section and conclusion section. Also, the description of “no large-scale survey’s…” was a mistake and deleted.

For the second major comment, we moved the previous studies description to the introduction so that the section of research methodology become clear. Then, we simplified the description in the results section. We also simplified the descriptions in the result section and eliminated some unnecessary information so that there is no repeated information. Then we added some descriptions regarding “trust” in the discussion section and tried to clear our new knowledge in addition to the previous studies. Finally, we tried to reflect back on the studies aims and objectives with the new knowledge in the conclusion.

Our responses to the further comments are as follows;

I revised some means of expression in the figures and diagrams. Some figures were deleted in accordance with the above mentioned deletion of description.

 I revised all the sentences written in the first person (anywhere).

My colleague checked the English.

This manuscript is a resubmission of an earlier submission. The following is a list of the peer review reports and author responses from that submission.

Round 1

Reviewer 1 Report

This manuscript uses online survey to identify determinants of farmers' and non-farmers' attitudes toward climate change adaptation in Japan. The paper finds that farmers tend to prefer "protection" and "insurance" as adaptive strategies, while non-farmers are more receptive to "withdrawal" than the farmers. The paper also links perceptions with trust and values in decision-making process. The topic is interesting and important, yet the current form of the paper does not live up to the standard to be published. I highlight my concerns as below: 

1) the biggest issues I found, is that the paper uses different sampling frame for farmers and non-farmers. Non-farmers are recruited, stratfied by gender and age groups, in three agrricultural prefectures, while farmers are recruited nationally. It is not clear to me how these two groups can be properly compared. For example, the Yamanashi prefecture suffered from heavy snow in 2014, as noted on page 5, thus the non-farmers may be more awared of climate chaneg due to such weather shock. If farmers are not recruited in the same location, the comparison is in vain. This problem also make the results of total sample (N=1035) difficult to interpret. These 1000+ people do not represent the rural population in Japan. One way to mitiate this issue is to weight farmer/non-farmer in the results. However, even with weighting, the samples are still not representative due to the different sampling frame. In any case, the authors need to address this flaw forcifully. In the current manuscript, the authors only mention this issue in passing in the method section. This flaw is so serious that I suggest rejection on this ground. 

2) The authors can improve the presentation of the results. On page 5, the paper mentions the variable concerning "experience with disaster" and responses on "social capital," yet I cannot find such varables anywhere in the paper. If there is, it needs to be clarified. Also, it is not clear what the significanc level in Fig 2 and Fig 4 represents. If I am correct, there is no mention of such analysis in the paper. 

3) The authors seem to have information on the social demographic background of the respondents. I thought it would be interesting and useful to see how the results vary across these variables. The authors mentioned that the non-farmers are stratified by age groups and gender, but did not actually use age and gender. I suggest the authors add a section on this topic. 

4) I suggest the authors include the original survey instrument as an appendix. 

5) In the paper, I suggest the authors make it more clear that it is a story based in Japan. This means when referring the "Ministry," it should be Japan's Ministry, etc. They can also add a little more contexts on Japan's climate adpatation efforts to orient international audiences. 

Reviewer 2 Report

The introduction could provide more references.

Section 2 "Previous studies and research methods" should be reviewed: 

- some references could be moved to the introduction;

- provide a more detailed description of the study area (total area, agricultural area, characteristics of farms, ...);

- provide a more detailed description of survey.

The figures must be reviewed as some numbers are not readable. Figure 6 is very complex, it should be simplified for better readability. 

Reviewer 3 Report

Review of Baba and Tanaka Climate Jan 2019

This paper describes the results of a survey of over 1000 Japanese people living in rural areas (both farmers and non-farmers) that addressed their attitudes towards climate change adaptation methods. The study has some interesting results and I believe is it a worthwile addition to the literature, once the comments below have been addressed.

Major suggestions

·         I’m a bit concerned about the premise of the paper that climate change is already impacting people's lives in a significant way. Yes, this is probably true, but saying that all heavy rainfall or snow events are climate change is a bit misleading. I suggest that you revise the paper to soften the language here, or provide supporting references on the attribution of climate change on recent extreme events in Japan

·         It's not clear why you want to 'increase the preference' for withdrawal, as mentioned in the conclusion. Can you provide some explanation for this in the introduction or the conclusion?

·         The paper would benefit from a proof-read of a native English speaker. I have identified some errors below, but there are likely others

·         It would be good to see an expansion of table 1 or even a map providing more information about the survey participants to accompany the description in section 2. Can you be explicit about the number of participants in each of the 18 categories? Also, what efforts were put in place to ensure the participants details remained secure?

·         It would also be good for the survey questions to be published with the study. This could be done as a supplementary section, or the exact questions that are discussed could be provided in figure captions.

Minor comments 

Line 14: serious what?

Line 18: you could remove the words 'is that', and replace them with a comma

Line 20: remove the word 'an' before agricultural

Line 41: please specify that the ministry is Japanese

Line 51: 'terms' not 'term'

Line 58–60: this sentence does not make sense to me

Line 68: 'through a large-sizes collective data so far' does not make sense to me

Line 89–92: does not makes sense to me

Line 93: I think you need to explicitly state that the 'adults of both sexes' are a sample of non-farming participants. At the moment the sentence reads as though you are only talking about the participants who are farmers.

Line 170: can you provide an example of the agricultural-related adaptation method you prepared for the farmers?

Line 235: Can you please define Cronbach's alpha?

Line 240: Please spell out the acronyms AGFI, CFI and RMSEA

Line 263–270: This section is a bit confusing and needs to be revised. Also, where is your evidence that farmers are worried about the lack of successors and international/national circumstances?

Figure 1: 'lifestyle' rather than 'lifestile'

Figure 2 and Figure 5: change the 'P<0.01**' and 'P<0.05*' to '** = P < 0.01' and '* = P < 0.05'. It is currently confusing. Also please expand the caption to explain how these thresholds are defined

Author Response

Major suggestions

1.        I’m a bit concerned about the premise of the paper that climate change is already impacting people's lives in a significant way. Yes, this is probably true, but saying that all heavy rainfall or snow events are climate change is a bit misleading. I suggest that you revise the paper to soften the language here, or provide supporting references on the attribution of climate change on recent extreme events in Japan

→ I tried to soften the language but couldn't find the proper word. I think the added survey questions according to your suggestion on No. 5 would stress that the description is based on the feeling and perception of respondents, that is, all heavy rainfall or snow events do not result from climate change necessarily in terms of scientific evidence (e.g. Line 142-143).

2.        It's not clear why you want to 'increase the preference' for withdrawal, as mentioned in the conclusion. Can you provide some explanation for this in the introduction or the conclusion?

→ I deleted this phrase because it does not required necessarily. (Line 366)

3.         The paper would benefit from a proof-read of a native English speaker. I have identified some errors below, but there are likely others

→ Actually, the paper have already benefited from a proof-read of a native English speaker, but I found some more errors other than the below you indicated. I fixed them everywhere in the paper. 

4.         It would be good to see an expansion of table 1 or even a map providing more information about the survey participants to accompany the description in section 2. Can you be explicit about the number of participants in each of the 18 categories? Also, what efforts were put in place to ensure the participants details remained secure?

→ I clarified them in the following place (Line 124-125, table 1, 130-131).

5.         It would also be good for the survey questions to be published with the study. This could be done as a supplementary section, or the exact questions that are discussed could be provided in figure captions.

→ I added the exact questions not in figure caption but in the body text (Line 142-143, 192-193, 206-208, 217-218).

Minor comments 

Line 14: serious what? → I revised the sentence (Line 13-14).

Line 18: you could remove the words 'is that', and replace them with a comma → (Line 18)

Line 20: remove the word 'an' before agricultural → (Line 21)

Line 41: please specify that the ministry is Japanese → (Line 41)

Line 51: 'terms' not 'term' → (Line 54)

Line 58–60: this sentence does not make sense to me → I revised the sentence (Line 57-60)

Line 68: 'through a large-sizes collective data so far' does not make sense to me → I revised the sentence (Line 71).

Line 89–92: does not makes sense to me → I revised the sentence (Line 89-91).

Line 93: I think you need to explicitly state that the 'adults of both sexes' are a sample of non-farming participants. At the moment the sentence reads as though you are only talking about the participants who are farmers. → I revised the sentence (Line 96).

Line 170: can you provide an example of the agricultural-related adaptation method you prepared for the farmers? → I added the sentence (Line 220-229).

Line 235: Can you please define Cronbach's alpha? → I added the sentence (Line 285-286)

Line 240: Please spell out the acronyms AGFI, CFI and RMSEA → I added the words (Line 291-292).

Line 263–270: This section is a bit confusing and needs to be revised. Also, where is your evidence that farmers are worried about the lack of successors and international/national circumstances? → I deleted the confusing sentence and revised some words (Line 316-321).

Figure 1: 'lifestyle' rather than 'lifestile' → (Figure 1)

Figure 2 and Figure 5: change the 'P<0.01**' and 'P<0.05*' to '** = P < 0.01' and '* = P < 0.05'. It is currently confusing. Also please expand the caption to explain how these thresholds are defined → (Figure 2 and Figure 5)
